# Large-Scale Coastal Marine Wildlife Monitoring with Aerial Imagery

**DOI:** 10.3390/jimaging11040094

**Published:** 2025-03-24

**Authors:** Octavio Ascagorta, María Débora Pollicelli, Francisco Ramiro Iaconis, Elena Eder, Mathías Vázquez-Sano, Claudio Delrieux

**Affiliations:** 1Departamento de Ingeniería, Universidad Nacional de la Patagonia San Juan Bosco, Puerto Madryn 9120, Argentina; ascagorta.octavio@gmail.com (O.A.); pollicelli@cenpat-conicet.gob.ar (M.D.P.); 2Departamento de Física, Instituto de Física del Sur, Universidad Nacional del Sur (UNS) and CONICET, Bahía Blanca 8000, Argentina; francisco.iaconis@uns.edu.ar; 3Centro para el Estudio de Sistemas Marinos, Centro Nacional Patagonico, CONICET, Puerto Madryn 9120, Argentina; eder@cenpat-conicet.gob.ar; 4Departamento de Biologia, Universidad Nacional de Catamarca, San Fernando del Valle de Catamarca 4700, Argentina; vazquezmathias696@gmail.com; 5Departamento de Ingeniería Eléctrica y Computadoras, Instituto de Ciencias e Ingeniería de la Computación, Universidad Nacional del Sur and CONICET, Bahía Blanca 8000, Argentina

**Keywords:** coastal marine wildlife monitoring, environmental management, aerial imagery, deep learning, population assessment

## Abstract

Monitoring coastal marine wildlife is crucial for biodiversity conservation, environmental management, and sustainable utilization of tourism-related natural assets. Conducting in situ censuses and population studies in extensive and remote marine habitats often faces logistical constraints, necessitating the adoption of advanced technologies to enhance the efficiency and accuracy of monitoring efforts. This study investigates the utilization of aerial imagery and deep learning methodologies for the automated detection, classification, and enumeration of marine-coastal species. A comprehensive dataset of high-resolution images, captured by drones and aircrafts over southern elephant seal (*Mirounga leonina*) and South American sea lion (*Otaria flavescens*) colonies in the Valdés Peninsula, Patagonia, Argentina, was curated and annotated. Using this annotated dataset, a deep learning framework was developed and trained to identify and classify individual animals. The resulting model may help produce automated, accurate population metrics that support the analysis of ecological dynamics. The resulting model achieved F1 scores of between 0.7 and 0.9, depending on the type of individual. Among its contributions, this methodology provided essential insights into the impacts of emergent threats, such as the outbreak of the highly pathogenic avian influenza virus H5N1 during the 2023 austral spring season, which caused significant mortality in these species.

## 1. Introduction

Monitoring coastal marine species whose colonies span large areas (e.g., over a few hundred kilometers of coastline) poses significant challenges stemming from the vastness, remoteness, and inaccessibility of these habitats, which render traditional on-site censuses logistically demanding and resource-intensive [1]. Compounding these difficulties are environmental variability, seasonal migrations, and human disturbances, all of which hinder the accurate collection of data essential for biodiversity conservation and ecosystem management [2]. Conventional methods for in situ wildlife censuses and monitoring have several limitations and drawbacks, particularly when compared to advanced technologies such as aerial surveys [3]. Remote and hazardous locations often make fieldwork both risky and costly, while on-site efforts can cause disturbance to habitats or introduce invasive species [4]. Moreover, traditional sampling techniques, such as simple random sampling, are ill-suited for dynamic and mobile populations, leading to inconsistent estimates and trends [5,6,7]. Ground counts, in particular, suffer from inaccuracies due to individual occlusion and the dense packing of animals in colonies, often resulting in significant underestimations of population sizes [4].

The use of aerial surveys has great potential to mitigate these limitations, giving rise to an influx of research initiatives in recent years [8]. Aerial surveys offer enhanced precision, yet they necessitate the development of standardized and scalable approaches to fully leverage their potential for accurate and efficient biodiversity monitoring. Manned aerial surveys are performed routinely in the evaluation of fauna distribution and density over large geographic areas [9]. A key advantage of this technology is the ability to cover large areas and entire colonies in significantly shorter times compared to ground surveys, allowing for accurate population estimates without errors induced by animal movements at the sites. The use of high-resolution cameras relieves the observers from the duty of performing the actual counting during the flight. Compared to alternative monitoring methods, aerial imagery offers distinct advantages. Satellite imagery, while useful for large-scale habitat mapping, often lacks the spatial resolution required to identify individual animals or distinguish between species in densely populated colonies. Ground-based surveys, on the other hand, are limited by their logistical complexity, high costs, and potential to disturb wildlife and habitats. In contrast, aerial surveys strike a balance between coverage and detail, enabling the monitoring of vast areas with sufficient resolution to detect and classify individual animals. However, the use of manned aircrafts has some disadvantages, including their environmental impact and the fact that imagery takes for coastal surveys in areas with geographical features like high cliffs are often oblique (i.e., prone to errors induced by occlusion and animal density). Additionally, the subsequent work and human effort required to analyze images in the lab, along with the associated infrastructure costs, are significant drawbacks [10].

Unmanned aircraft systems are currently augmenting or replacing manned aerial surveys [11]. In [12], a drone-based mammal monitoring methodology is presented, specifically focusing on identifying factors affecting the detection reliability of drone images and understanding the effects of movement on the detection certainty in field trials in Northern Norway, surveying humpback and killer whales and harbor porpoises. In [13], the authors evaluated the use of drones to optimize the monitoring of marine fauna stranding on beaches in the Garopaba region of Brazil. The study compared drone-based surveys with traditional land-based monitoring methods. The results indicated that drones could significantly reduce the time required for monitoring beaches and potentially cause less disturbance to the animals present on the beach compared to the use of land vehicles or boats. A specific challenge in drone-based monitoring is related to megafauna assemblages. In [14], the authors found that megafaunal assemblages differed significantly among beaches in the northern, central, and southern regions of New South Wales, Australia. The models explaining most of the variation in megafaunal assemblages included bird exposure and water temperature. A key limitation of the study was that the data were limited to wave-exposed beaches and may not be generalizable to other coastal environments. A comprehensive overview of the current applications and challenges of using drones in marine mammal research is presented in [15]. This review analyzed 169 publications and found that the main uses included abundance and distribution monitoring, photo-identification, morphometric estimates, blow sample collection, and behavioral studies.

Drones offer several advantages for coastal wildlife monitoring, including the availability of very high-resolution cameras and the ability to capture nadir-oriented images. These features help mitigate errors caused by lighting conditions (e.g., shadows cast by cliffs or individuals) and by the dense crowding of individuals. Even though drones may be successful as a non-invasive and useful tool, there are several limitations related to autonomy, generalizability of monitoring, data processing time, and regulatory restrictions. Also, it strongly relies on weather conditions for safe operation, requires previous permits from the application authorities for flights beyond the extended visual line of sight, and the surveyed area is usually smaller and restricted by the equipment’s battery life (at least when using low-cost rotary-wing drones). The use of lithium batteries also poses an environmental impact.

Aerial surveys, both manned and unmanned, introduce a new challenge: processing the large volumes of high-resolution imagery generated, each containing tens to hundreds of individuals. This necessitates time-intensive and laborious human-assisted assessment, highlighting the need for automated approaches to streamline image analysis. In this context, supervised computer vision models based on deep learning architectures appear as a clear alternative [16]. This strategy is advantageous in the sense that having a large number of images, each with potentially several hundreds of animal instances, grants sufficient training data [17]. However, the raw images are not adequate for training, since a previous supervised annotation stage is required [18]. Distinguishing between animal species often requires expert assistance in the annotation or validation procedures, as individuals of certain species or sex and age (male, female, juvenile, etc.) may appear very similar to the untrained eye [19]. Moreover, numerous species exhibit natural camouflage or mimicry, allowing them to blend with their surroundings and thereby posing additional challenges for accurate identification. Additionally, the appearance of marine animals can vary significantly regarding their breeding or reproductive cycles, introducing variability in ground-truth annotations [20]. Also, there are variances and uncertainties introduced by variability arising during the takes (flight altitude, oblique camera angles, illumination conditions, etc.), which require contextual cues for proper annotation.

Even with a robust annotation procedure and little label noise, deep learning models face significant limitations in wildlife detection, particularly when dealing with species that exhibit morphological similarities, camouflage, or complex environmental backgrounds. For instance, these models often struggle to distinguish between visually similar species, such as different deer or bird species, due to overlapping features [21,22]. Additionally, animals with natural camouflage or those partially obscured in their habitats pose challenges, as models may fail to detect them accurately [23]. Variability in lighting, camera angles, and animal behavior further complicates detection, leading to reduced model performance [24,25]. Inaccuracies in population estimates, such as underestimations due to animal occlusion or dense packing in colonies, can lead to flawed conservation strategies and mismanagement of natural resources [26]. For example, in southern elephant seals, accurate monitoring is essential to understand long-term population dynamics, site fidelity, and dispersal patterns [27,28,29]. Underestimations can distort interpretations of survival, reproduction, and habitat use, undermining conservation efforts. They can also misrepresent genetic structure and demographic history, leading to incorrect assumptions about recovery from population declines. Such errors result in misguided hypotheses about species’ responses to environmental changes, exacerbating the vulnerability of threatened species and ecosystems. These limitations highlight the need for more robust and adaptable deep learning approaches to improve wildlife monitoring and biodiversity conservation efforts.

In this paper, we present a model that automates the survey and population assessment of wildlife in coastal natural areas of Northern Patagonia using image processing and deep learning techniques. The available information consists of high-resolution images captured during two transect flights conducted within less than six hours along a 300 km stretch of the coastal littoral of the Península Valdés. These flights and the gathered imagery encompassed the entire breeding colony of elephant seals, several haul-out colonies of sea lions, and other species and objects of interest. A subset of this imagery was used to develop and train an image analysis model designed to detect, identify, and survey marine wildlife targets of interest in the region (initially focusing on elephant seals and sea lions, given the specific situation to be mentioned below) according to the demands of different stakeholders (research groups, regulatory authorities, non-governmental organizations, etc.). The flights and information gathering were indeed timely due to the contingency caused by the Highly Pathogenic Avian Influenza (HPAIV H5N1) outbreak that provoked unprecedented mortality in both species during the last spring season in October 2023, causing important, potentially long-term ecological impacts. The results of this project will unify the tasks of data collection and analysis processes across different species, enabling the implementation of strategies for containing and preventing infectious diseases as well as mitigating the impacts of environmental threats. These tools will be essential for the continuous monitoring of marine population health in this ecologically critical region, where maintaining systematic, long-term monitoring is essential for assessing the population status of several marine species. Additionally, they will provide the possibility to expand the monitoring efforts to include a wider range of species and environmental elements in the future.

## 2. Materials, Model Building, and Evaluation

The images for this study were obtained during an aerial survey conducted aboard a single-engine, high-wing CESSNA C-182 aircraft. The total surveyed site covered a stretch of more than 300 km of coastline in the northern part of Chubut Province, Patagonia, Argentina, from Punta Buenos Aires to Punta Tombo (see Figure 1). This area includes the entire breeding colony of Southern Elephant Seals (*Mirounga leonina*, SES) [30] and several stable or occasional haul-out colonies of South American Sea Lions (*Otaria flavescens*, SL) [31], as well as other wildlife species and objects of interest. The images used for the fauna identification model (*n* = 461) encompass approximately 80 km of this surveyed area, covering the external coastal front of Península Valdés from Punta Delgada to Punta Norte. This section provides a comprehensive representation of the heterogeneity of the colonies and their surrounding spatial environment. The survey was conducted during the austral spring, on 4 October 2023, coinciding with the peak of the SES breeding season, in an unprecedented context for the area due to the massive outbreak of highly pathogenic avian influenza (HPAI H5N1), which severely affected individuals of both species.

A manned flight was preferred for two reasons: first, to conduct a comprehensive demographic survey of the entire SES colony, which requires simultaneous imaging (or with minimal temporal difference) across the entire area. Second, the use of drones, depending on the location, may require access to private properties, where landowners have the right to deny entry, making coastal access uncertain. Nonetheless, drone imaging was conducted in locations where it was feasible, with the future goal of training a generative model to convert aerial images into their drone-equivalent counterparts. During the flight, the minimum altitude and minimum distance from the coast were approximately 150 m, and the flight speed was approximately 165 km/h, during which a set of oblique aerial images were taken with a SONY ILCE-7RM10 camera. Simultaneously, a mobile georeferencing application (GPS logger) was used to georeference the images. The camera generates timestamp metadata when taking each image. A script was developed to align this timestamp metadata with the metadata of the GPS logger, which provided accurate spatiotemporal registration. This procedure is unnecessary in cameras with built-in GPS.

For the study presented here, we used a subset of 461 images. The image resolution was reasonably high (7952×5304), but in most takes, the actual size of the animals occupied only a small fraction of the whole frame (see Figure 2). Thirteen annotators manually labeled the images after receiving personalized training to ensure consistent identification and annotation. Some annotators also participated in pre-aerial monitoring ground censuses, enhancing their contextual understanding. A detailed annotation guide with examples was provided, and periodic virtual meetings were held for feedback and clarification. Annotation was performed using the LabelStudio tool (https://labelstud.io/, accessed on 21 February 2025), labeling regions of interest (ROIs) as bounding boxes. The initial labeling process included 44 classes, comprising southern elephant seal (SES) individuals categorized by sex, age, state (alive or dead), and social context, South American sea lion (SL) individuals (alive or dead), seabird (SB) individuals of different species, and other relevant objects commonly found in the habitat. The adopted labeling format follows a *superclass_class_subclass* schema (see Figure 3). When the annotators had completed their annotation sessions, specialized biologists reviewed the annotations for mistakes or confusion.

The focus of this research is on coastal marine fauna monitoring, specifically in SES population assessment, and evaluation of the impact of the pandemic. For this reason, we retained only the most relevant 19 classes, grouped and renamed into seven coarse-grained superclasses (see Table 1). Given that LabelStudio does not support hierarchical labeling (i.e., taxonomies), we opted for a labeling schema that may allow for future model retraining and refinement using a flexible subclass–superclass grouping. The grouping criteria considered not only relevance but also morphological similarity, visual distinction among species or categories, and the number of labeled instances per class to ensure a more balanced distribution across superclasses.

An additional class *confounding* was also added, and several thousand confounding contacts were labeled to test the false positive rate of the model. These confounding contacts included objects or image features (e.g., rocks, shadows, unrelated fauna) that may affect the detection. In some cases, the confounding contacts may have been actual objects of interest, and for this reason, two different labels (overt and dubious) were included. In Figure 4, an instance of each of these superclasses is presented.

To optimize the model training process, the labeled dataset was preprocessed using the following pipeline. First, the dataset of 461 images was exported in COCO format and processed using the SAHI (Slicing Aided Hyper Inference) library (https://github.com/obss/sahi, accessed on 21 February 2025). Using SAHI, the images were divided into 640×640 tiles—the default resolution in most pre-trained computer vision architectures—as this size offers a good balance between detection accuracy and computational efficiency. A 10% overlap was applied between adjacent tiles to prevent missed detections along tile borders [32]. This process generated a total of 81,136 tiles.

Then, a filtering step was applied to the tiles. For this purpose, an ad hoc algorithm was designed to filter out irrelevant tiles by analyzing the presence of bounding boxes associated with the defined target classes as well as their instances balance. Tiles that did not contain regions of interest (ROIs) corresponding to the selected classes were discarded. This included tiles with only background or content from other classes. Additionally, to balance the instances of the predominant bird sub-class (seagull), tiles containing only seagull ROIs were also excluded. This filtering process reduced the total number of tiles to 10,306. Finally, the dataset was converted into YOLO format for training.

These tiles were then split into training, validation, and test sets using an 80%–10%–10% ratio, ensuring an even class distribution across all three subsets. Dataset splitting was performed using the scikit-learn library (https://scikit-learn.org, accessed on 21 February 2025). To prevent data leakage—particularly given the high similarity between sequential aerial images—we implemented a temporal grouping strategy. Specifically, image tiles derived from the same original image or from images captured within a 10 s interval were randomly assigned to the same subset (training, validation, or test). This ensured that no overlapping or highly similar images were present in both the training and test sets, thereby maintaining the integrity of our evaluation metrics. Although the split was randomized, this temporal grouping strategy effectively prevented the model from being exposed to highly similar data during training and testing, which could otherwise lead to artificially inflated performance metrics.

Although the number of instances in the majority classes was sufficient for effective model training, data augmentation (described in Table 2) was applied to enhance the model’s robustness and improve its ability to generalize to new instances. Figure 5 presents the complete workflow.

Taking into account computational efficiency, previous work in similar applications [34], and available documentation, YOLO (https://www.ultralytics.com/yolo, accessed on 21 February 2025) was chosen for model training. YOLO architectures are notorious for employing a single-stage backbone, as opposed to the two-stage backbones in other architectures. Since its initial proposal in 2015, it has undergone several iterations and improvements, becoming one of the most popular object detection frameworks to date [35]. Of the available YOLO releases, several trials were performed with YOLO v8n, YOLO v8l, and YOLO v10x (see Appendix A). Unlike previous versions of YOLO, YOLO v10x incorporates a more robust detection head and enhanced feature extraction mechanisms, which improve its ability to detect small objects in complex backgrounds. This is particularly important for our study, where animals often appear at varying scales and angles due to the oblique nature of the aerial imagery. Additionally, YOLO v10x’s optimized architecture strikes a balance between accuracy and computational efficiency, making it well-suited for large-scale monitoring tasks that require processing high-resolution images with minimal latency. Considering an adequate trade-off between model accuracy and computational efficiency, we finally opted for YOLO v10x. The dataset exhibits a significant class imbalance, with some classes (e.g., sea birds) having substantially more annotations than others (e.g., male SES). To mitigate the impact of this imbalance on model performance, we incorporated class weights into the loss function during training. The weight for each class is $w_{j}=\frac{n_{s}}{n_{c}\times n_{j}}$, where $n_{s}$ is the total number of samples in the dataset, $n_{c}$ is the total number of unique classes, and $n_{j}$ is the number of samples for the respective class *j*. This weighting scheme ensures that the loss function assigns higher importance to instances of underrepresented classes, thereby encouraging the model to learn more effectively from these classes. By incorporating class weights, we aimed to reduce the bias toward overrepresented classes and improve the overall generalization of the model.

## 3. Results

The setting configuration of the model architecture parameters and training hyperparameters is presented in Table 3. Hyperparameter tuning is mainly automated by the development library. The only setting that needs adjustment is the batch size, which depends on available hardware. In our implementation of YOLO v10x, we employed a combination of loss functions to optimize object detection performance. The box loss ensured precise bounding box predictions, the classification loss (cls_loss) improved class prediction accuracy, and the distribution focal loss (dfl_loss) refined the bounding box distribution for enhanced localization precision. Figure 6 shows these values together with the precision and recall for both the training and validation datasets across training epochs.

The confusion matrix is shown in Figure 7. In addition to the seven superclasses mentioned above, we included an extra row and column to assess overall false positives (false alarms) and false negatives (misses). The bottom row, labeled “false detection”, includes the false positives for each superclass raised by the model, i.e., instances where the model mistakenly detects a specific contact in the background. The rightmost column, “miss”, counts the amount of misses per superclass, i.e., the number of labeled examples of each superclass that was not detected by the model. It can be observed that the female SES, suckling SES, male SES, and sea bird superclasses achieved precision rates close to or above 80%, with the first two standing out for their individual prediction percentages. On the other hand, the weaned SES and sea lion superclasses achieved precisions of 68% and 77%, respectively. Table 4 presents the precision, recall, F1 score, and mAP metrics calculated from the confusion matrix data.

The average precision (AP) and mean average precision (mAP) at 50% and 95%, as well as the intersection over union (IoU) thresholds, which are the most widespread evaluation metrics [36], were also computed. Figure 8 shows the precision–recall curve for the selected model and individual classes. When applying the mentioned test, a mAP of 0.78 was achieved, with an AP of 0.96 for the female SES superclass, an AP of 0.89 for the suckling SES superclass, and an AP of 0.86 for the male SES superclass. The sea bird superclass was identified with an AP of 0.79, and the sea lion superclass had an AP of 0.70. The weaned SES superclass had an AP value of 0.63.

All training and evaluation procedures were performed using a Intel Core i0-13900KF processor-based desktop computer with 32 G DDR5 RAM and a Nvidia RTX 3080TI 12Gb GDDR6 GPU. The training time was about six minutes per epoch (i.e., ten hours for a 100 epoch cycle). Model inference was be performed per tile (given the resolution limitations of the YOLO architecture) in about 30 ms per tile. This final feature means that once the model was trained, the complete unfiltered flight dataset (81,136 tiles) could be processed in less than an hour.

## 4. Discussion

The 2023 breeding season was atypical, marked by significant changes in the SES social dynamics. In historically high-density areas, few reproductive males leading harems were observed, and groups of females with pups were often found alone [37]. This suggests that alpha adult males—typically older, larger, and stronger individuals that dominate hierarchical fights to access mating opportunities within harems [30]—may have been atypically underrepresented during 2023. Based on this possibility, misclassifications between males and females may have involved sub-adult males (usually subordinated and peripheral males) due to their smaller size and less-pronounced features. However, we cannot confirm this definitively, as accurately distinguishing between age categories of sub-adult males in aerial imagery is challenging. Environmental factors such as poor lighting or occlusions likely contributed to these errors, as the model struggled to distinguish subtle differences in size and morphology under such conditions. These inaccuracies can distort interpretations of survival, reproduction, habitat use, and genetic structure, ultimately undermining conservation efforts and misrepresenting species’ responses to environmental changes. Over time, these inaccuracies can cascade through the ecosystem, misrepresenting overall biodiversity resilience. By addressing these challenges, our study aims to contribute to more accurate surveillance that supports ecosystem stability and biodiversity preservation. Continued monitoring policies (e.g., periodical flights) with automated population estimation will certainly reduce the uncertainties regarding the assessment of ecologic dynamics and provide support for more informed conservation practices.

The developed model achieved a mean average precision (mAP) of 0.78, with particularly high average precision (AP) scores for SES superclasses such as female SES (0.95) and suckling SES (0.92). Also, it is remarkable that the model attained a high accuracy for the male SES superclass, notwithstanding its very low prevalence. This very low prevalence is expected under normal conditions due to the longer time it takes for males to reach sexual maturity and actively participate in breeding seasons. On the contrary, the weaned SES class (also with very low prevalence, since most of the nursing pups did not survive as a result of the epidemic) achieved a comparatively lower precision among the superclasses. The actual factors that reduce precision and recall are due to false positives and negatives with respect to the background (i.e., false detections and misses) and not to mutual confusion.

The sole exception observed was that approximately 12% (13 out of 108) of male elephant seals were misclassified as females. This misclassification can be attributed to the primary basis of sex distinction among adult SES individuals, which relies predominantly on sexual dimorphism, specifically differences in size and the morphological characteristics of the head. Since size cannot be accurately assessed in uncalibrated images and the facial features indicative of males may be obscured or not clearly visible in the images, the model lacks the necessary cues to correctly identify the individual’s sex.

The sea bird and sea lion superclasses (which were not in our initial scope) still performed well (precisions of 0.82 and 0.77, respectively), and most misclassifications were false positives or false negatives (i.e., very low confusion with other superclasses), suggesting the feasibility of using the imagery and retraining the model for the population dynamics of these coastal species. Finally, the behavior of the model with respect to confounding contacts also showed classification robustness. The low recall of this superclass (0.58) was mostly due to misses, and only very few cases of confusion among superclasses arose. Similarly, the lack of precision in this superclass was mostly due to false positives rather than superclass confusion.

All these aspects of the dataset and the model imply that with proper retraining, the results can be reused for different purposes (monitoring of other coastal species or training models for more specific classification within classes). This last alternative (e.g., identifying dead animals) is specifically relevant given the outbreak of highly pathogenic avian influenza virus (HPAIV H5N1) that caused significant mortality in both southern elephant seals and sea lions. The capacity to accurately and quickly assess population metrics using the developed tools will help in understanding the impacts of emergent threats such as this and in implementing strategies for preventing infectious diseases. The annotation process demonstrated a robust approach to handling complex ecological imagery. The hierarchical labeling schema captured detailed target information, essential for ecological monitoring, in a flexible and practical manner. Training sessions with the annotators, a detailed guide, and iterative feedback ensured consistency and accuracy of the labeling process. Annotators with field experience further improved reliability, while real-time metric monitoring maintained high annotation quality.

## 5. Conclusions

Manned aerial transects are able to cover large areas quickly and efficiently. This enables the collection of accurate population estimates without the errors induced by animal movements at the sites. The use of high-resolution cameras also eliminates the need for manual counting, reducing human effort and potential biases. However, this approach introduces challenges in image annotation, particularly the reliance on expert assistance to distinguish between similar-looking species, sex, and age categories, as well as careful assisted interpretation of confounding contacts.

We presented a robust and scalable methodology for large-scale coastal marine wildlife monitoring using aerial imagery and deep learning. Our findings highlight the potential of these techniques to improve population assessment, facilitate ecological studies, inform conservation efforts, and demonstrate the effectiveness of using aerial imagery and deep learning for the automated detection, classification, and enumeration of coastal marine wildlife, specifically focusing on southern elephant seals and South American sea lions in the Valdés Peninsula, Argentina. The use of a deep learning framework, particularly the YOLO v10x architecture, enabled the accurate assessment of population metrics from high-resolution manned aerial images, achieving overall F1 scores of between 0.7 and 0.9. These results highlight the potential of this method to improve population assessments, facilitate ecological studies, and inform conservation efforts. It also offers significant advantages over traditional in situ methods (including drone takes), which are often limited by logistic constraints, environmental variability, and potential disturbances to wildlife.

The primary advantages of our approach include its scalability, efficiency, and ability to process large datasets with minimal human intervention. It also reduces disturbances to wildlife compared to traditional in situ methods, including drone surveys, which can be logistically challenging, limited by the equipment’s autonomy (usually less than 7–8 km for the most accessible models under ideal conditions), and susceptible to environmental variability. However, this method is not without limitations. The reliance on high-resolution aerial imagery introduces potential biases, such as variability in image quality due to flight altitude, oblique camera angles, and the presence of confounding elements like shadows or rocks in the background. Additionally, the generalization of the model to other species or environments may require further training and validation.

This study represents an advancement in wildlife monitoring by combining manned aerial imagery with state-of-the-art deep learning techniques. The development of a multi-class system that is capable of identifying species, sex, and age categories and can flexibly adapt to different contexts can be considered a step forward compared to previous studies. To the best of our knowledge, this is the first work to apply such a framework for the simultaneous detection and classification of multiple wildlife targets in coastal marine environments, specifically in this particular geographical region and these animal species.

Our results and methodology may shed light on the specific context of the HPAIV H5N1 outbreak. It has been hypothesized [37] that this outbreak could have disproportionately impacted breeding male SES, potentially increasing their mortality rates due to prolonged virus exposure. This is supported by the atypical number of dead males observed during the 2023 breeding season. While these hypotheses are beyond the scope of this study, our methodology significantly facilitates continued monitoring. A new manned flight with images processed using the approach proposed in this study will provide the data needed to rapidly analyze population responses in future generations and determine the required parameters for population dynamic studies with more precision. This integration of AI-based monitoring into conservation strategies will demonstrate the significant potential for long-term biodiversity preservation.

Our results can be further improved, refined, and expanded, offering a powerful tool for biodiversity monitoring and environmental management. In particular, this study represents a first step toward developing a multi-class system capable of identifying each animal subcategory. Among the venues for future research, we demonstrated the potential of expanding the model to include a wider range of species and environmental elements. Also, it is feasible to train generative models to convert aerial images into drone-equivalent counterparts, which could make animal identification and classification easier and more accurate. Further investigation into novel image detection architectures and specific data augmentation techniques is also worthwhile, particularly those that address the variance introduced by take variabilities like flight altitude, oblique camera angles, and the presence of confounding elements in the background (stones or shadows). Future work could also explore semi-automated tools and expanded field-based training to enhance efficiency and contextual understanding. Overall, this framework provides a strong foundation for wildlife assessment using aerial imagery and deep learning.

## Figures and Tables

**Figure 1 jimaging-11-00094-f001:**
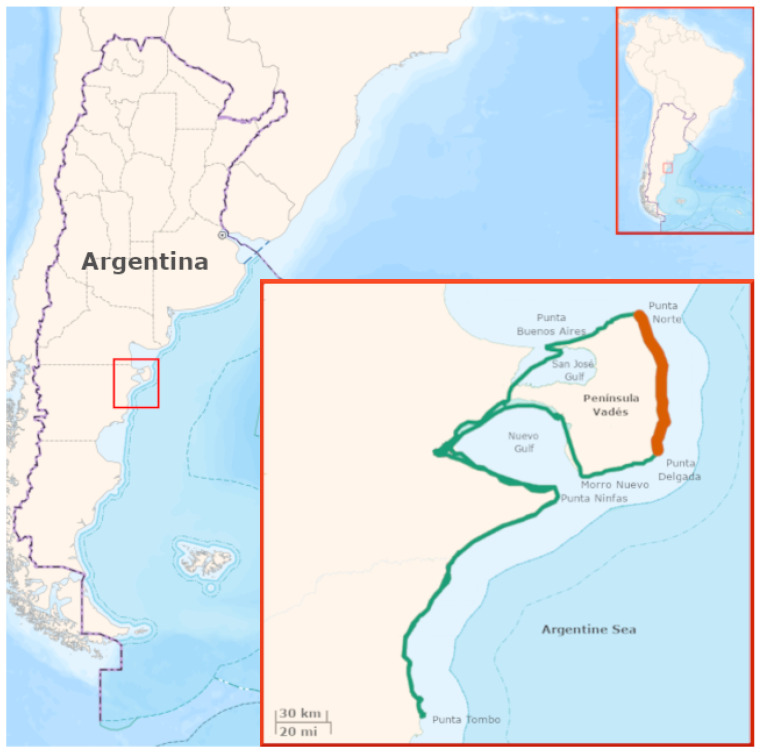
Aerial survey along the coast of Northern Patagonia, Argentina (green), focusing on the Península Valdés sector (the transect where images were captured for the deep learning training subset is shown in red).

**Figure 2 jimaging-11-00094-f002:**
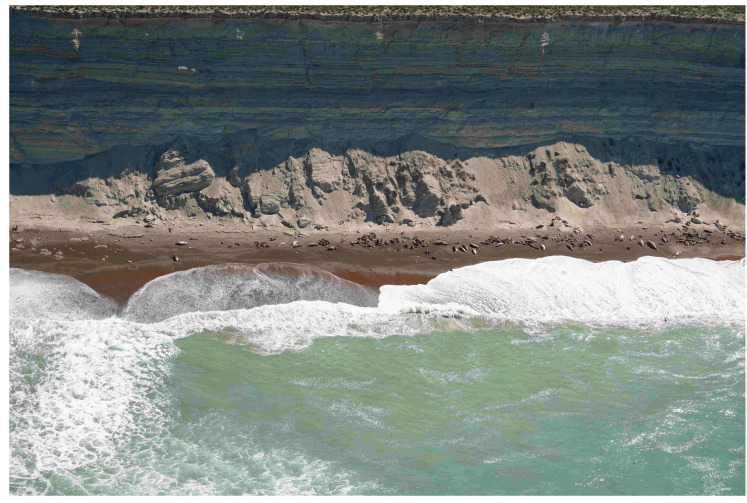
Image in the Punta Delgada-Punta Norte sector (Península Valdés).

**Figure 3 jimaging-11-00094-f003:**
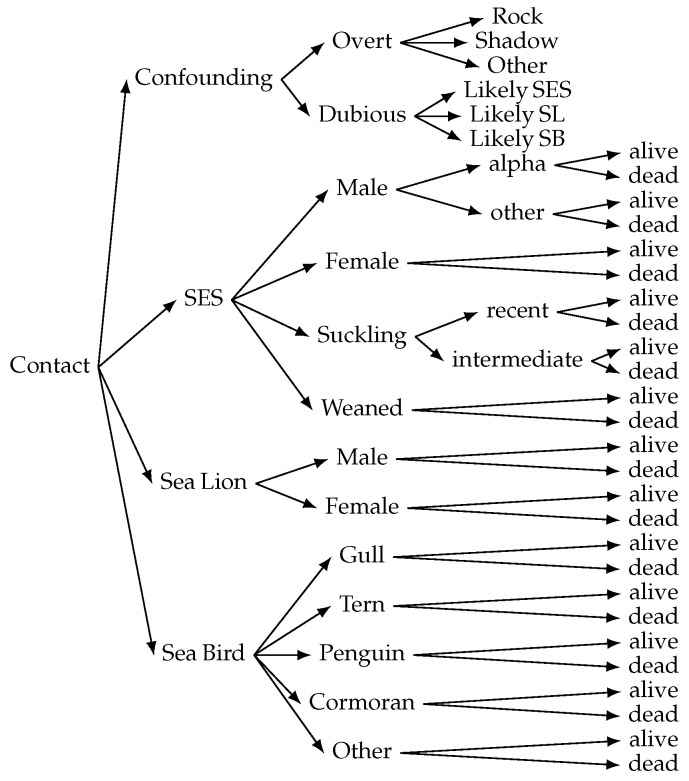
Labeling taxonomy of the contacts.

**Figure 4 jimaging-11-00094-f004:**
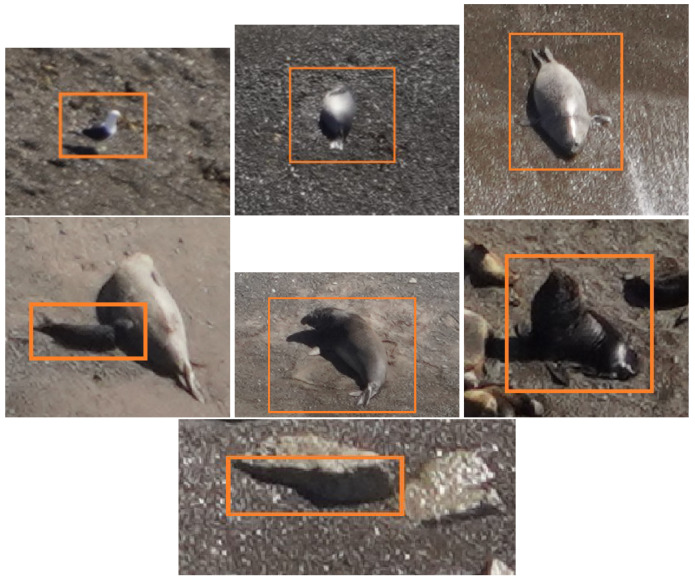
Instances of the labeled detections (**left** to **right**, **top** to **bottom**): Sea bird, weaned SES, female SES, suckling SES, male SES, sea lion, confounding contact. These instances are shown in the original resolution as crops of the original images to show that in many cases, the actual size of the targets occupy only a few pixels.

**Figure 5 jimaging-11-00094-f005:**
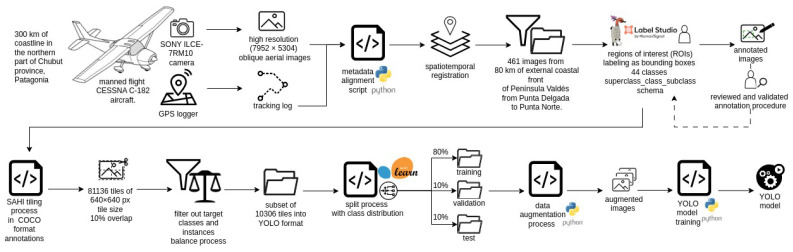
Block diagram showing the complete processing workflow.

**Figure 6 jimaging-11-00094-f006:**
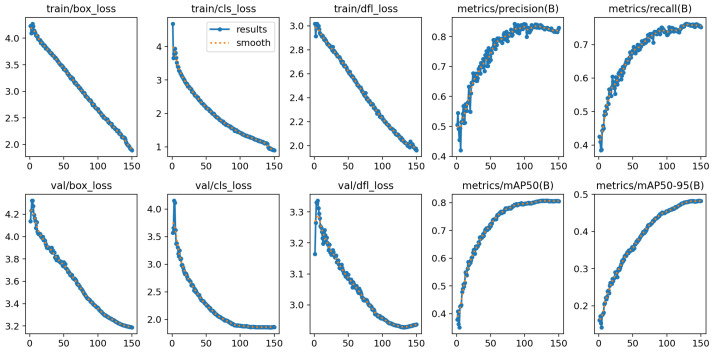
YOLO v10x after 150 epochs.

**Figure 7 jimaging-11-00094-f007:**
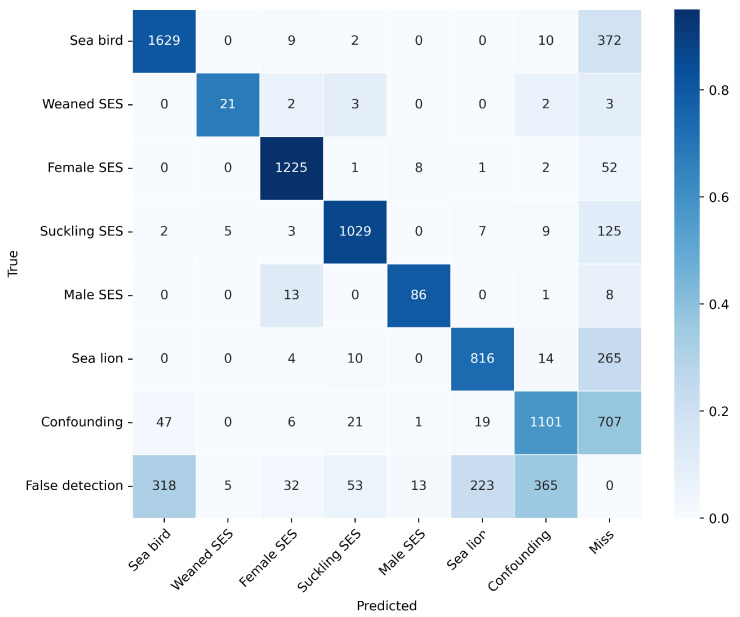
Confusion matrix of the model (color scale represents per-class prevalence).

**Figure 8 jimaging-11-00094-f008:**
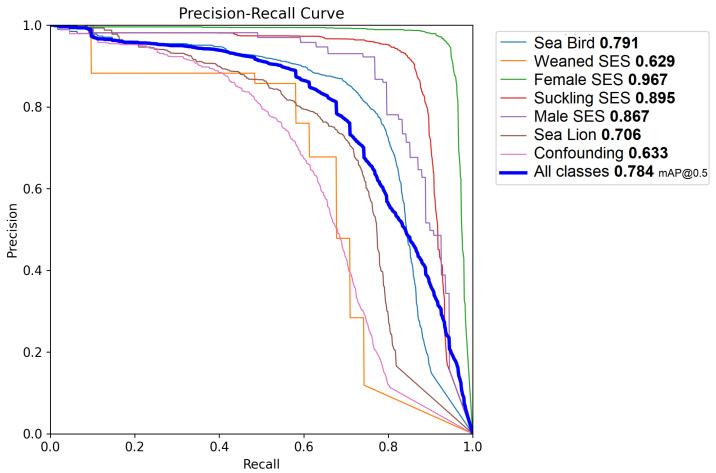
Per-class precision–recall curves and mAP values.

**Table 1 jimaging-11-00094-t001:** Labeling classes, grouped by superclasses, and amount of instances per class.

Superclass	Class	Instances
Sea Bird (SB)	SB_cormoran	281
	SB_sea-gull	18.962
	SB_tern	4
	SB_other	379
	SB_penguin	5.676
Weaned SES (WSES)	WSES_alive	327
	WSES_dead	29
Female SES (FSES)	FSES_alive	12.988
	FSES_dead	64
Suckling SES (SSES)	SSES_intermediate_alive	2.920
	SSES_intermediate_dead	647
	SSES_recent_alive	2.920
	SSES_recent_dead	5.966
Male SES (MSES)	MSES_alpha_alive	596
	MSES_alpha_dead	55
	MSES_other_alive	274
	MSES_other_dead	54
Sea Lion (SL)	SL_alive	10.825
	SL_dead	1.608
Confounding (C)	C_overt	13.537
	C_dubious	6.077

**Table 2 jimaging-11-00094-t002:** Applied data augmentation transformations.

Transformation	Value	Description
fliplr	0.5	Horizontal flip with 50% probability.
hsv_v	0.4	Random brightness variation up to 40%.
hsv_s	0.7	Random saturation variation up to 70%.
hsv_h	0.015	Random hue variation up to 1.5%.
mosaic	1.0	Combines four images into one for training.
scale	0.5	Random scaling up to 50% of the image size.
crop_fraction	1.0	Random cropping of up to 100% of the image.
auto_augment	randaugment	Applies random augmentations (e.g., rotation, distortion) [33].

**Table 3 jimaging-11-00094-t003:** Model architecture and training hyperparameters.

Category	Parameter	Description	Value
Model architecture	model	Model version	yolov10x
	nc	Number of classes	7
	backbone	Backbone network	CSPNet
	neck	neck additional features	Path aggregation network
	anchors	Predefined anchors for detection	Default
Training hyperparameters	epochs	Number of training epochs	150
	batch	Batch size (images per batch)	9
	imgsz	Input image size	640
	optimizer	Optimization algorithm	SGD
	lr0	Initial learning rate	0.01
	momentum	Momentum for optimizer	0.937
	weight_decay	L2 regularization	0.0005
	warmup_epochs	Number of warm-up epochs	3.0
	warmup_momentum	Momentum during warm-up	0.8
	warmup_bias_lr	Learning rate for biases in warm-up	0.1
	box	Weight for bounding box loss	7.5
	cls	Weight for classification loss	0.5

**Table 4 jimaging-11-00094-t004:** Evaluation metrics.

Class	Precision	Recall	F1 Score	mAP
Sea bird	0.82	0.81	0.81	0.79
Weaned SES	0.68	0.68	0.68	0.62
Female SES	0.95	0.95	0.95	0.98
Suckling SES	0.92	0.87	0.90	0.90
Male SES	0.80	0.80	0.80	0.88
Sea lion	0.77	0.74	0.75	0.71
Confounding	0.73	0.58	0.64	0.63

## Data Availability

The original contributions presented in this study are included in the article. Further inquiries can be directed to the corresponding author.

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
