# Peer review of "Large-Scale Coastal Marine Wildlife Monitoring with Aerial Imagery"

_2313-433X, 2025, doi:10.3390/jimaging11040094_

Round 1

Reviewer 1 Report

Comments and Suggestions for Authors

Author Response

The article presents a novel approach to monitoring coastal-marine wildlife using aerial imagery combined with deep learning techniques. While the use of aerial surveys for ecological monitoring is not entirely new, the application of a deep learning framework for automated species identification and population assessment in marine environments represents a significant advancement. The integration of high-resolution drone and aircraft imagery with AI-driven analysis enhances efficiency and accuracy, making this study a valuable contribution to the field. The introduction provides sufficient background information and includes relevant references. It effectively outlines the challenges of coastal-marine wildlife monitoring and the advantages of using aerial surveys and AI-based techniques.The article is well-structured and clearly written. However, some sections could benefit from a more streamlined presentation, particularly in the methodology and results sections, to improve clarity and reduce redundancy.

R1.1: We appreciate the reviewer's judgment of our contribution, and will try to address all their comments in what follows.

For example: The description of image acquisition and data processing is very detailed, which is useful but somewhat repetitive. For example, explaining the use of SAHI (Slicing Aided Hyper Inference) and the dataset splitting process is valuable, but some steps are described at length when they could be summarized more succinctly.

R1.2: Thanks for raising this aspect. We understand the concern and rewrote these descriptions in a more succinct way. However the other three reviewers are asking further details and therefore it is difficult to arrive into a consensual writing.

The discussion on different YOLO versions tested before selecting YOLOv10x is interesting, but presenting a direct justification for the final choice instead of detailing multiple trials might make the section clearer.

R1.3: Thanks for this remark. We added a paragraph explaining the advantages of this particular architecture (lines 241-260), and added in the appendix some experiments showing the different results that justified the choice.

The results are clearly presented, with relevant figures and tables supporting the findings. The confusion matrix and precision-recall curves provide transparency regarding model performance. The topic is of high interest to researchers in ecology, conservation biology, remote sensing, and artificial intelligence. The interdisciplinary approach broadens its appeal, making it relevant to both scientific and applied fields. The study’s findings have practical implications for real-world conservation efforts, increasing its relevance beyond academia.

R1.4: We sincerely appreciate the reviewer's comments.

Overall Merit: The article makes a significant contribution to the field of ecological monitoring by demonstrating the effectiveness of combining aerial imagery with deep learning for wildlife assessment. The methodology is robust, and the findings have important implications for conservation efforts. Minor improvements in presentation and additional discussion on annotation reliability could further strengthen the study. Overall, this is a well-executed and impactful piece of research.

R1.6: We expanded the methods section to detail in more depth the annotation procedure we implemented to grant reliability (lines179-191). Also in the discussion section we highlighted this aspect of our research (lines 353-358)

Reviewer 2 Report

Comments and Suggestions for Authors

The authors' team has proposed a manuscript on monitoring coastal wildlife, especially counting marine-coastal species using deep learning methods. It describes a novel application of deep learning to automate wildlife monitoring and fits the journal's scope. As a reviewer, I´d like to conclude that the article's topic is exciting and actual, and I´ve "read it at once."

Unfortunately, despite its attractiveness to the readers, it has to be rewritten to be "more scientific." In its current form, I´m finding it more like a conference topic than a current contents article. A lot of the processes are explained by two or three sentences, thus very simply without more profound analysis. The reader is only informed that something was done and has to believe that it was done correctly or that the results are acceptable; thus, the manuscript lacks the meaning of reproducibility.

From the formal point of view, the manuscript is well written and generally interesting to the readers. It deals with an actual topic, and as it concerns wildlife and endangered species monitoring, it has also added social value. The length of the article is adequate, the chapters are in logical order, the research was conducted properly, and the results are interpreted correctly. The English language is fine; only some minor misspellings have to be corrected, which I believe will be done by the MDPI Editor during proofreading.

The authors should include more references to other teams or methods in the Introduction section to compare and prove that their approach to the research is correct and that their way of solving the tasks/problems is innovative and relevant to the study.

The Materials chapter will benefit from a block diagram showing how the images were obtained, processed, segmented, and evaluated and how the model was trained. The manuscript lacks any mathematical backing.

The conclusion section should be more specific about the results and summarise the pros and cons of this method, the accuracy of the results, and what is unique about this manuscript. Also, the limitations of the potential biases in aerial imagery and generalization of the model should be added.

Summary: The manuscript is actual, fits the journal scope, and is very interesting to readers. It uses actual methods and solves actual problems. After revision, I´d recommend it for acceptance for publication.

Author Response

The authors' team has proposed a manuscript on monitoring coastal wildlife, especially counting marine-coastal species using deep learning methods. It describes a novel application of deep learning to automate wildlife monitoring and fits the journal's scope. As a reviewer, I´d like to conclude that the article's topic is exciting and actual, and I´ve "read it at once."

R2.1: We are sincerely grateful for the reviewer's comments.

Unfortunately, despite its attractiveness to the readers, it has to be rewritten to be "more scientific." In its current form, I´m finding it more like a conference topic than a current contents article. A lot of the processes are explained by two or three sentences, thus very simply without more profound analysis. The reader is only informed that something was done and has to believe that it was done correctly or that the results are acceptable; thus, the manuscript lacks the meaning of reproducibility.

R2.2: Thank you for your positive feedback. We made an effort in the revised version to describe  the procedures in a more parsimonious way, we provided additional references to prior work for further context, and ensured that the main results are explicitly supported and clearly presented in tables and figures throughout the manuscript. In particular, we expanded the methods and the discussion sections in the revised manuscript to detail the steps taken to ensure consistent annotations (see lines lines179-191 and lines 353-358), we added a more thorough justification of the model's architecture (lines 241-260), and provided a more extensive error analysis and interpretation of the results (lines 328-335) . We hope these adjustments address these concerns and provide greater clarity. For the sake of reproducibility, we added a block diagram with a parsimonious description of all the steps in the processing workflow (Figure 5) detailing the tasks and the software frameworks used at each. Finally, we added an ablation study showing the effect of data augmentation, and a statistical analysis of the confidence intervals of the model (both in the appendix).

From the formal point of view, the manuscript is well written and generally interesting to the readers. It deals with an actual topic, and as it concerns wildlife and endangered species monitoring, it has also added social value. The length of the article is adequate, the chapters are in logical order, the research was conducted properly, and the results are interpreted correctly. The English language is fine; only some minor misspellings have to be corrected, which I believe will be done by the MDPI Editor during proofreading.

R2.3: Thanks for your positive comments. We have carefully reviewed the text to correct any minor misspellings or grammatical issues regarding the English language.

The authors should include more references to other teams or methods in the Introduction section to compare and prove that their approach to the research is correct and that their way of solving the tasks/problems is innovative and relevant to the study.

R2.4: We have added additional citations to works by other teams and methods that align with or contrast our research in order to better contextualize our approach, and provide additional relevance of our results (lines 105-124). Additionally, we would like to emphasize that, to the best of our knowledge, there are no previous studies that have utilized this type of tool to simultaneously detect and classify multiple wildlife targets. This further underscores the uniqueness and relevance of our approach.

The Materials chapter will benefit from a block diagram showing how the images were obtained, processed, segmented, and evaluated and how the model was trained. The manuscript lacks any mathematical backing.

R2.5: Thanks for pointing this out. As mentioned above, we added a new figure with the complete workflow (Fig. 5 ). As per the mathematical backing, the underlying theoretical framework of deep learning models is presented in a good amount of freely accessible sources, so in our opinion it wouldn't add anything to the manuscript, making it only unnecessary longer.

The conclusion section should be more specific about the results and summarise the pros and cons of this method, the accuracy of the results, and what is unique about this manuscript. Also, the limitations of the potential biases in aerial imagery and generalization of the model should be added.

R2.6: We rewrote the conclusion section to meet the observations, adding several paragraphs (lines 363-366, lines 373-408, lines 416-422) to address the reviewer's remarks.

Summary: The manuscript is actual, fits the journal scope, and is very interesting to readers. It uses actual methods and solves actual problems. After revision, I´d recommend it for acceptance for publication.

R2.7: We appreciate the reviewer's comment and judgement, and hope the new version of the manuscript addresses all their concerns to satisfaction.

Reviewer 3 Report

Comments and Suggestions for Authors

This manuscript presents a method for large-scale coastal-marine wildlife monitoring using aerial imagery and deep learning, which is of great significance for biodiversity conservation and environmental management. The authors collected high-resolution images, developed and trained a deep learning model, and achieved notable results in the detection and classification of coastal-marine species. However, during the review process, several areas for improvement were identified.

1.The background introduction could be further strengthened. Although the manuscript mentions the challenges of monitoring coastal-marine species, it does not delve into the specific ecological implications of these challenges. For instance, it does not clearly explain how the inaccuracies in data collection directly affect the understanding of ecosystem balance and the long-term impacts on biodiversity. Additionally, the literature review could be expanded, such as by explicitly addressing the limitations of existing deep learning models in wildlife detection (e.g., handling species with morphological similarities or camouflage), which would better highlight the innovation of this study.

2.It is suggested to further increase the latest research literature in the recent three years, especially in 2025. e.g. “Adaptive Downsampling and Scale Enhanced Detection Head for Tiny Object Detection in Remote Sensing Image”,“An Efficient Perceptual Video Compression Scheme Based on Deep Learning-Assisted Video Saliency and Just Noticeable Distortion”. The concepts of these papers share similarities with your methodology.

3.The manuscript provides a detailed description of data collection, image preprocessing, and model training processes, but lacks rigorous justification for some key technical choices. For example, the rationale for selecting YOLOv10x is not sufficiently explained; it only mentions a trade-off between accuracy and computational efficiency. A comparative analysis of this architecture’s unique advantages (e.g., handling oblique imagery or dense animal clusters) over alternatives (e.g., YOLOv8, Faster R-CNN) would strengthen the methodology.

4.The experimental section is comprehensive but lacks direct comparisons with state-of-the-art methods. It is recommended that the authors further explore the methodological advantages of their approach in large-scale wildlife monitoring. Incorporating ablation studies or performance comparisons with baseline models would better demonstrate the contributions of individual components to the overall system.

  1. The article provides visual display of some test results, but lacks detailed analysis and display of specific test results
Comments on the Quality of English Language

The English could be improved to more clearly express the research.

Author Response

This manuscript presents a method for large-scale coastal-marine wildlife monitoring using aerial imagery and deep learning, which is of great significance for biodiversity conservation and environmental management. The authors collected high-resolution images, developed and trained a deep learning model, and achieved notable results in the detection and classification of coastal-marine species. However, during the review process, several areas for improvement were identified.

R3.1: We appreciate the reviewer's comments and will try to address all their concerns in the new manuscript.

1.The background introduction could be further strengthened. Although the manuscript mentions the challenges of monitoring coastal-marine species, it does not delve into the specific ecological implications of these challenges. For instance, it does not clearly explain how the inaccuracies in data collection directly affect the understanding of ecosystem balance and the long-term impacts on biodiversity. Additionally, the literature review could be expanded, such as by explicitly addressing the limitations of existing deep learning models in wildlife detection (e.g., handling species with morphological similarities or camouflage), which would better highlight the innovation of this study.

R3.2: Thanks for pointing out these issues. We expanded the literature review in the introduction to include references that address the limitations of existing deep learning models in wildlife detection. Also, we included the ecological implications of monitoring challenges for coastal-marine species, including a detailed explanation of how inaccuracies in population estimates can directly impact species like southern elephant seals (lines 105-124).

2.It is suggested to further increase the latest research literature in the recent three years, especially in 2025. e.g. “Adaptive Downsampling and Scale Enhanced Detection Head for Tiny Object Detection in Remote Sensing Image”,“An Efficient Perceptual Video Compression Scheme Based on Deep Learning-Assisted Video Saliency and Just Noticeable Distortion”. The concepts of these papers share similarities with your methodology.

R3.3: We appreciate these suggestions. Some of the most recent literature is unavailable for research groups in Argentina, given the dreadful cutoffs imposed by the current government to scientific research (see for instance the recent reports in Nature https://www.nature.com/articles/d41586-024-03994-y and Science https://www.science.org/content/article/scienticide-argentina-s-science-workforce-shrinks-government-pursues-austerity). We are citing and reviewing only the references that we were actually able to find.

3.The manuscript provides a detailed description of data collection, image preprocessing, and model training processes, but lacks rigorous justification for some key technical choices. For example, the rationale for selecting YOLOv10x is not sufficiently explained; it only mentions a trade-off between accuracy and computational efficiency. A comparative analysis of this architecture’s unique advantages (e.g., handling oblique imagery or dense animal clusters) over alternatives (e.g., YOLOv8, Faster R-CNN) would strengthen the methodology.

R3.4: Thanks for this remark. We added a paragraph explaining the advantages of this particular architecture (lines 241-260), and added an appendix showing the different results that justified the choice.

4.The experimental section is comprehensive but lacks direct comparisons with state-of-the-art methods. It is recommended that the authors further explore the methodological advantages of their approach in large-scale wildlife monitoring. Incorporating ablation studies or performance comparisons with baseline models would better demonstrate the contributions of individual components to the overall system.

R3.5: Thanks for this remark. We included in the appendix an ablation study showing how the data augmentation methodology increased the model performance. With minor variations, our framework is the widespread one adopted in the relevant bibliography that we cited, but to the best of our knowledge has never been applied to marine-coastal fauna in this specific geographical region.

5. The article provides visual display of some test results, but lacks detailed analysis and display of specific test results.

R3.6: Thanks for pointing out this aspect. We expanded the discussion section to include comments on error analysis and results validation (lines 328-335) and added a figure in the appendix (Figure 11) with the display of specific test results.

Reviewer 4 Report

Comments and Suggestions for Authors

COMMENTS

The authors propose a deep learning-based approach for the automated detection, classification, and enumeration of coastal-marine wildlife using aerial imagery. The study focuses on monitoring southern elephant seals and South American sea lions in the Valdés Peninsula, Patagonia, Argentina. The methodology combines high-resolution imagery captured by drones and aircraft with YOLOv10x-based deep learning models to enhance wildlife monitoring accuracy. The study highlights the impact of the 2023 outbreak of Highly Pathogenic Avian Influenza (HPAIV H5N1) on these species and proposes a scalable solution for long-term ecological surveillance. While the work is innovative and demonstrates strong technical contributions, there are areas where clarity and depth can be improved. Below are detailed comments followed by a final recommendation.

Major Concerns:

  • The authors claim that the proposed YOLOv10x model is effective, but there is no comparison with alternative models (e.g., Faster R-CNN, EfficientDet, or other YOLO versions). The authors should include baseline results from a few other deep learning models for a fair assessment of performance.
  • The manuscript states that multiple versions of YOLO were tested (YOLOv8n, YOLOv8l, and YOLOv10x), but it does not provide performance metrics for these models. A table or graph comparing mAP, precision, recall, and F1-score across tested architectures should be included to justify the choice of YOLOv10x.
  • The dataset appears highly imbalanced (e.g., ~19,000 sea bird annotations vs. a few hundred for weaned SES). The authors should provide an in-depth discussion on how this imbalance affects model performance. Resampling techniques or weighted loss functions should be considered to mitigate class imbalance.
  • The authors mention an 80%-10%-10% train-validation-test split but do not clarify whether this was randomized or stratified by class distribution. It is essential to discuss whether data leakage was prevented, especially given the high similarity between sequential aerial images.
  • The results primarily rely on mAP and precision-recall curves, but there is no statistical analysis of model performance. The authors should provide confidence intervals for mAP and statistical significance testing (e.g., Wilcoxon signed-rank test or bootstrapping) to validate results.
  • The manuscript applies various augmentation techniques (e.g., flipping, HSV transformations, mosaic augmentation), but it does not assess their individual impact.
  • An ablation study should be conducted to determine which augmentations contribute most to performance improvements.
  • The model’s failure cases (e.g., confusion between male and female SES) should be analyzed in greater depth. The authors should discuss how environmental factors (e.g., lighting variations, occlusions, water reflections) affect detection accuracy.
  • While the study presents a large-scale aerial monitoring approach, the authors do not discuss inference speed (i.e., how long it takes to process an image). A comparison of processing times for different models and hardware setups would improve practical applicability.

Minor Concerns:

Abstract & Introduction

  • The abstract is well-structured, but it does not mention key performance results (e.g., mAP, precision, recall).
  • Introduction should explicitly contrast aerial imagery with alternative monitoring methods (e.g., satellite imagery, ground-based surveys).

Methodology Section

  • The georeferencing method used to align GPS metadata with image timestamps should be explained in greater detail.
  • Hyperparameter tuning strategy (learning rate schedules, regularization techniques) should be detailed.

Results & Discussion

  • The confusion matrix is useful, but misclassifications should be analyzed in more detail (e.g., explaining why males were misclassified as females).
  • Discussion should provide more ecological insights on the implications of AI-based monitoring for long-term biodiversity conservation.

Figures & Tables

  • Some figures are low resolution and difficult to interpret.
  • Table 4 should include standard deviations or confidence intervals for precision, recall, and F1-score.

Some minor points

  1. Avoid using abbreviations and acronyms in titles, abstract, and headings.
  2. The first time you use an acronym in the text, please write the full name and the acronym in parenthesis. Do not use acronyms in titles, abstract, and chapter headings.
  3. Also, the list of references should be carefully checked to ensure consistency between all references and their compliance with the journal policy on referencing.
  4. There is a need for language improvement. I found some grammatical error texts in the manuscript. The language of the paper needs review.

Overall Assessment:

The manuscript provides a promising contribution to AI-based wildlife monitoring but requires substantial improvements in experimental validation, result interpretation, and discussion of limitations. Recommendation: Major Revision.

Comments on the Quality of English Language

The English could be improved to more clearly express the research.

Author Response

The authors propose a deep learning-based approach for the automated detection, classification, and enumeration of coastal-marine wildlife using aerial imagery. The study focuses on monitoring southern elephant seals and South American sea lions in the Valdés Peninsula, Patagonia, Argentina. The methodology combines high-resolution imagery captured by drones and aircraft with YOLOv10x-based deep learning models to enhance wildlife monitoring accuracy. The study highlights the impact of the 2023 outbreak of Highly Pathogenic Avian Influenza (HPAIV H5N1) on these species and proposes a scalable solution for long-term ecological surveillance. While the work is innovative and demonstrates strong technical contributions, there are areas where clarity and depth can be improved. Below are detailed comments followed by a final recommendation.

R4.1: We appreciate the reviewer's comments and will try to address all their concerns in the new manuscript.

Major Concerns:

  • The authors claim that the proposed YOLOv10x model is effective, but there is no comparison with alternative models (e.g., Faster R-CNN, EfficientDet, or other YOLO versions). The authors should include baseline results from a few other deep learning models for a fair assessment of performance.

R4.2: Thanks for this remark. We added a paragraph explaining the advantages of this particular architecture (lines 241-260), and added in the appendix some experiments showing the different results that justified the choice. We acknowledge that there are currently many other backbone options for image detection, and indeed there is plenty of space for fine-tuning custom architectures. However, this would slow down the obtention of good enough results which are of the essence in the current environmental situation. We added this assessment as further research in the conclusion section (lines 416-422).

  • The manuscript states that multiple versions of YOLO were tested (YOLOv8n, YOLOv8l, and YOLOv10x), but it does not provide performance metrics for these models. A table or graph comparing mAP, precision, recall, and F1-score across tested architectures should be included to justify the choice of YOLOv10x.

R4.3: In the appendix mentioned above we included a comparison showing that our architecture choice appears to be the most effective.

  • The dataset appears highly imbalanced (e.g., ~19,000 sea bird annotations vs. a few hundred for weaned SES). The authors should provide an in-depth discussion on how this imbalance affects model performance. Resampling techniques or weighted loss functions should be considered to mitigate class imbalance.

R4.4: Thanks for pointing out this omission. We added a description of the weighted loss function used in the model in lines(251-260).

  • The authors mention an 80%-10%-10% train-validation-test split but do not clarify whether this was randomized or stratified by class distribution. It is essential to discuss whether data leakage was prevented, especially given the high similarity between sequential aerial images.

R4.5: Thanks again for remarking on another omission. We added an explanation on the specific type of random train-validation-test split that was adopted (lines 224-232).

  • The results primarily rely on mAP and precision-recall curves, but there is no statistical analysis of model performance. The authors should provide confidence intervals for mAP and statistical significance testing (e.g., Wilcoxon signed-rank test or bootstrapping) to validate results.

R4.6: Thanks for this suggestion. We performed a statistical analysis of the model performance using the multiple test sets approach. The results are in the appendix.

  • The manuscript applies various augmentation techniques (e.g., flipping, HSV transformations, mosaic augmentation), but it does not assess their individual impact.

R4.7: Assessing the individual impact of each augmentation technique would require retraining the model afresh for each one, which would take much longer time than the established deadline for the new manuscript (roughly 12 to 15 hours per training cycle). Moreover, these experiments are not worth much since the combined effects of each augmentation technique don't add up. An extensive study, thus, would imply an exploration of the combinatorial space of augmentations. 

  • An ablation study should be conducted to determine which augmentations contribute most to performance improvements.

R4.8: In response to this and to the previous remark, we conducted a "small" ablation study just dropping all augmentations, to show how the actual model performance was enhanced by the combined action of all of them. The results can be found in the Appendix.

  • The model’s failure cases (e.g., confusion between male and female SES) should be analyzed in greater depth. The authors should discuss how environmental factors (e.g., lighting variations, occlusions, water reflections) affect detection accuracy.

R4.9: Thanks for raising this issue. As is mentioned in the manuscript (lines 328-335) the most relevant (both in statistical and biological terms) are the 13 cases of male SES classified as female SES, which represent approximately 11% of false negatives. These cases involve male SES where the key sexual dimorphism feature -the proboscis or trunk on the face, located on the head- is not clearly visible due to occlusions or self-occlusions. This fact also explains why there are no inverse cases. The features and meaning of the failure cases were already mentioned in the original manuscript, but a paragraph clarifying this issue and the consequences on the use of the model was added in the new manuscript (lines 299-318).

  • While the study presents a large-scale aerial monitoring approach, the authors do not discuss inference speed (i.e., how long it takes to process an image). A comparison of processing times for different models and hardware setups would improve practical applicability.

R4.10: Thanks for this suggestion, we added a line in the results section with this and related information (lines 291-297)

Minor Concerns:

Abstract & Introduction

  • The abstract is well-structured, but it does not mention key performance results (e.g., mAP, precision, recall).
  • Introduction should explicitly contrast aerial imagery with alternative monitoring methods (e.g., satellite imagery, ground-based surveys).

R4.11: Thanks for making these points. We included performance results in the abstract, and expanded the comparison of aerial imagery wrt alternative monitoring methods already present in the introduction (lines 43-50).

Methodology Section

  • The georeferencing method used to align GPS metadata with image timestamps should be explained in greater detail.
  • Hyperparameter tuning strategy (learning rate schedules, regularization techniques) should be detailed.

R4.12: Thanks for raising these issues. A more detailed description of the georeferencing is included (lines 173-176), but this is likely to become obsolete given that newer cameras already integrate GPS registration. Regarding hyperparameter tuning, we expanded the description (lines 263-265) mentioning that this kind of training features are automated within the deep learning frameworks.

Results & Discussion

  • The confusion matrix is useful, but misclassifications should be analyzed in more detail (e.g., explaining why males were misclassified as females).
  • Discussion should provide more ecological insights on the implications of AI-based monitoring for long-term biodiversity conservation.

R4.13: Please see reply 4.9.

Figures & Tables

  • Some figures are low resolution and difficult to interpret.
  • Table 4 should include standard deviations or confidence intervals for precision, recall, and F1-score.

R4.14: The only low-resolution images are the sample contacts in Figure 4, which, indeed, are the actual resolution of the contacts and RoIs (we added this in the caption of this figure). 

Some minor points

  1. Avoid using abbreviations and acronyms in titles, abstract, and headings.
  2. The first time you use an acronym in the text, please write the full name and the acronym in parenthesis. Do not use acronyms in titles, abstract, and chapter headings.
  3. Also, the list of references should be carefully checked to ensure consistency between all references and their compliance with the journal policy on referencing.
  4. There is a need for language improvement. I found some grammatical error texts in the manuscript. The language of the paper needs review.

R4.15: 1. The only abbreviation we found in abstract and headings is HPAIV H5N1 in the abstract, we omitted the initial acronym but H5N1 is the actual denomination of the strain. 2. We carefully checked to verify that no acronyms were used prior to its introduction. Also, we checked again that no acronyms are present in any heading. 3. We checked the list of references and found no miscompliance to the journal policy. 4. We carefully revised the text of the new manuscript to check grammar and typos.

Round 2

Reviewer 2 Report

Comments and Suggestions for Authors

Dear authors' team, Thank you for providing the revised version of the manuscript. It was improved based on the previous suggestions, and all of the concerns were addressed. I recommend accepting it in its current form.

Author Response

Dear authors' team, Thank you for providing the revised version of the manuscript. It was improved based on the previous suggestions, and all of the concerns were addressed. I recommend accepting it in its current form.

We are grateful for the reviewer's comments and suggestions that served to enhance the quality and significance of our work.

Reviewer 3 Report

Comments and Suggestions for Authors

The authors have carefully revised their manuscript according to my comments and suggestions. I have no other problems.

Comments on the Quality of English Language

 The English could be improved to more clearly express the research.

Author Response

The authors have carefully revised their manuscript according to my comments and suggestions. I have no other problems.

We sincerely appreciate the reviewer's comments and suggestions, which were essential for the quality of the final manuscript.

Reviewer 4 Report

Comments and Suggestions for Authors

The author has incorporated the suggested changes and improved revisions have been made in the paper and the revised version has the necessary qualities for acceptance compared to the previous version. The experimental results are good and well explained. In my opinion, the article is acceptable in its current form, and it can now be considered for publication.

Author Response

The author has incorporated the suggested changes and improved revisions have been made in the paper and the revised version has the necessary qualities for acceptance compared to the previous version. The experimental results are good and well explained. In my opinion, the article is acceptable in its current form, and it can now be considered for publication.

We are sincerely thankful for the comments, suggestions, and requests of the reviewer, which contributed significantly to the quality and significance of the final manuscript.